# Pipeline to evaluate YAP-TEAD inhibitors indicates TEAD inhibition represses *NF2*-mutant mesothelioma

Richard Cunningham[1], Siyang Jia[1], Krishna Purohit[1], Michaela Noskova Fairley[1], Marcin K Maniak[1], Yue Lin[1], Ning Sze Hui[1], Rebecca E Graham[2], Adriano G Rossi[1], Justyna Cholewa-Waclaw[3], Pierre O Bagnaninchi[4], Neil O Carragher[5], Carsten Gram Hansen[1]

As the core, tumorigenic downstream effectors of the Hippo signalling pathway, YAP/TAZ and the TEAD family of transcription factors represent attractive targets for drug discovery efforts within cancer research. This is particularly true in the context of pleural mesothelioma, in which there are many recent preclinical developments and clinical trials evaluating the efficacy of TEAD inhibitors. The range of inhibitors has shown great promise, but comparisons of their performances are so far limited. Here, we develop a high-content pipeline that enables a comparative analysis of currently developed YAP/TAZ-TEAD inhibitors. We take advantage of isogenic cellular models that enable us to examine inhibitor specificity. We identify genetic compensation of the Hippo pathway transcriptional module, with implications for therapeutic targeting, and implement Cell Painting to develop a detailed morphological profiling pipeline that enables further characterisation, quantification, and analysis of off-target effects. Our pipeline is scalable and allows us to establish specificity and comparative potency within cancer-relevant assays in a clinically relevant cellular model of pleural mesothelioma.

## Introduction

Pleural mesothelioma is an asbestos-induced cancer of the mesothelial lining of the lung and the deadliest cancer after diagnosis [1, 2]. Patients have a bleak outlook as many are diagnosed late or with aggressive disease, limiting the efficacy of surgical intervention [3, 4]. There are no curative therapies. To this end, therapeutics currently used can be considered palliative care and patients only survive on average for 12–18 mo after diagnosis. This highlights a clear and urgent need for the development of targeted therapies and implementation of effective chemotherapeutics [5].

As a relatively uncommon cancer type, pleural mesothelioma (PM) is defined by a lack of clear, high-penetrance driver mutations [6, 7, 8, 9, 10]. Despite this, PM is distinct from most common cancers in that genomic perturbations within upstream components of the Hippo pathway are overrepresented within its mutational profile [6].

The Hippo pathway comprises an upstream kinase cascade module, which functions to regulate the activity of the co-transcriptional activators YAP and TAZ [11, 12]. These, alongside the TEAD family of transcription factors [13, 14], represent the downstream transcriptional effectors of the Hippo pathway. Since the initial discovery of the signalling pathway as a core regulator of a variety of processes in development, differentiation, and regeneration [15, 16], YAP and TAZ have since been implicated as oncogenic drivers across a range of cancer types [17, 18, 19, 20, 21, 22, 23, 24]. Strikingly, PM is defined by a relatively high frequency of mutations within Hippo pathway relative to other cancer types [6, 7, 8, 9, 10]. These mutations include frequent loss-of-function mutations within the upstream Hippo pathway kinase cascade, including most frequently in *NF2*, with more infrequent loss-of-function mutations in *SAV1*, *LATS1*, and *LATS2* [6, 7, 8, 9, 10].

Although designing therapeutics against loss of tumour suppressors has been challenging historically, multiple inhibitors of the TEADs have recently been developed with the hopes to position for clinical use. The development of a range of varied direct inhibitors is an exciting advancement in cancer research, as hyperactive YAP/TAZ-TEAD activity is a widespread phenomenon across many cancers [25]. Most of the inhibitors developed comprise small molecule disruptors of TEAD autopalmitoylation, a post-translational modification required for the interaction between these transcription factors and YAP/TAZ [26, 27, 28]. After initial preclinical development, some of these promising inhibitors have now progressed to clinical trials, to be positioned as YAP/TAZ-TEAD inhibitors in PM patients. Because of the genomic evidence and frequent dysregulation of the Hippo pathway within PM, combined with both the ineffectiveness of surgery [29] and general

[1]Centre for Inflammation Research, Institute for Regeneration and Repair, The University of Edinburgh, BioQuarter, Edinburgh, UK    [2]Centre for Clinical Brain Sciences, Anne Rowling Regenerative Neurology Clinic, The University of Edinburgh, Edinburgh, UK    [3]High Content Screening Facility, Institute for Regeneration and Repair, The University of Edinburgh, Edinburgh, UK    [4]Centre for Regenerative Medicine, Institute for Regeneration and Repair, The University of Edinburgh, Edinburgh, UK    [5]Cancer Research UK Scotland Centre, Institute of Genetics and Cancer, The University of Edinburgh, Edinburgh, UK

Correspondence: Carsten.G.Hansen@ed.ac.uk

dearth of effective therapeutic interventions (10, 30), this cancer type is the primary focus in the development of these therapeutics.

Beyond selective inhibitors that are currently undergoing testing, there are a variety of indirect inhibitors of YAP/TAZ-TEAD. These comprise a heterogeneous range of chemicals, including dasatinib, a Src kinase inhibitor commonly used as an anticancer agent (31, 32), verteporfin, a photosensitiser historically used as a chemical inhibitor of YAP/TAZ activity either by directly disrupting YAP/TAZ-TEAD interaction (33, 34) or by sequestering YAP/TAZ in the cytoplasm via 14-3-3 proteins (35), and statins (36), a family of mevalonate pathway inhibitors reported to inhibit YAP/TAZ activity via the disruption of geranylgeranylation and downstream impact on F-actin polymerisation (19). We focus on these different therapeutic classes and prioritise a range of inhibitors with well-documented, potent anti-YAP/TAZ activity. This range of both selective and classical, less selective inhibitors is compared directly side by side, taking advantage of two separate and complementary isogenic cellular models to facilitate a direct therapeutic and molecular comparison within a clean, controlled genetic background.

# Results

## Identifying high potency inhibitors of YAP/TAZ-TEAD

With the current focus on YAP/TAZ-TEAD inhibition in the development of chemotherapeutics for use, especially for the treatment of PM (30, 37), three recently developed TEAD inhibitors were selected for testing within cell lines. These include VT-107 (38) and K-975 (39, 40), both pan-TEAD inhibitors, and IK-930, a TEAD1-specific inhibitor (41) that has undergone clinical trials (42) for the treatment of PM and other tumours associated with perturbations within the Hippo pathway (NCT05228015). A multitude of additional YAP-TAZ/TEAD inhibitors have in the past been identified through screening approaches and used primarily as tool compounds to probe preclinical impacts of Hippo pathway perturbations, as well as undergoing further preclinical development and optimisation (30, 43). Despite widespread use, most of these have less well-defined mechanisms of action (MoA) and might possess YAP-TAZ/TEAD inhibition as a secondary and potential off-target effect (34, 35). To compare the inhibitory potential of recently developed, on-target TEAD inhibitors with these more classic YAP inhibitors, we included verteporfin, the lipophilic statin lovastatin, and dasatinib as comparators (Fig 1A). Dasatinib is prioritised, as its primary mechanism of action as an anticancer therapeutic is via its function as a Src and tyrosine kinase inhibitor (31, 32), with dasatinib's highly potent downstream cellular impact on YAP/TAZ-TEAD appearing likely through the Src-YAP signalling axis (44, 45).

Initial testing was focused on the ability of compounds to inhibit cellular TEAD activity. This was analysed in HEK293A cells via the use of a luciferase reporter regulated by multiple TEAD binding sites (46). Quantification of TEAD activity upon 24-h treatment at 1 $\mu$M confirmed the ability of almost all tested compounds to inhibit TEAD activity (Fig 1B), apart from lovastatin. Interestingly, dasatinib exhibits the most profound inhibition of TEAD activity. To

validate this effect, we quantified the expression of YAP/TAZ signature genes, as defined by TCGA (47), on treatment with a selection of these inhibitors. This reveals a modest, though significant, decrease in signature expression (Fig S1A) under the same treatment conditions. Notably, the observed decrease in signature gene expression is TEAD-dependent, with no significant effects evident on treatment in TEAD KO cells (14) (Fig S1A). These data suggest that not only TEAD inhibitors, but also less specific inhibitors such as verteporfin, modulate the expression of YAP/TAZ target genes in a mainly TEAD-dependent manner. Given the difficulties associated in targeting YAP/TAZ directly because of their intrinsic disorder (30), these findings reinforce the likelihood that YAP/TAZ may be primarily targetable via their interaction with TEADs (30).

To probe the efficacy of candidate compounds within the context of mesothelioma, evaluation was conducted in a preclinical model of PM driver mutations, with CRISPR/Cas9-mediated knockout (48) (KO) of tumour suppressor NF2 in non-malignant, MeT-5A mesothelial cells. This unique isogenic cellular model represents the disease well (6) and is a powerful platform for interrogating YAP/TAZ-TEAD dynamics and responses to therapeutic inhibition. We have previously shown that the TCGA-defined YAP/TAZ signature gene set (47) is up-regulated upon mesothelial NF2 loss specifically upon cancer-relevant stresses (6). Initial testing focused on the treatment effect on YAP/TAZ-TEAD-mediated transcription via quantifying the expression of this target gene set. Computing the expression of this gene signature reveals that at 1 $\mu$M, most tested inhibitors are sufficient to down-regulate the transcription of the full gene set after 24-h treatment (Fig 1C), further validating their ability to suppress YAP/TAZ-TEAD activity. Interestingly, both IK-930, a TEAD1-specific inhibitor, and lovastatin did not decrease signature expression (Fig 1C), with the observed impact on YAP/TAZ signature expression generally matching the effect of treatment on TEAD activity as quantified by luciferase assays in HEK293A cells (Fig 1B).

YAP/TAZ-TEAD activity is contingent on nuclear retention of YAP/TAZ, with the upstream kinase component of the Hippo pathway known to phosphorylate YAP/TAZ, leading to subsequent cytoplasmic sequestration and therefore inhibition of these co-transcriptional activators (49, 50). To assess the dynamics of YAP/TAZ alongside the observed reduction in YAP/TAZ-TEAD-mediated transcription (Fig 1B and C), we next quantified the candidate compounds' inhibitory potential across a range of concentrations in the context of YAP nuclear localisation. YAP was selected as the sole readout of activity, with cellular TAZ dynamics omitted from study because of YAP and not TAZ being the prime determinant of tumorigenicity within the MeT-5A cellular model (6). In addition, as inhibitors are presumed to be cytostatic/cytotoxic and YAP nuclear localisation is reduced under conditions of high cell density (30, 51, 52), cells were filtered to those approximating 50% cell–cell contact to limit effects from contact inhibition at either extreme (6, 52). Although lower potency analogues of VT-107 and K-975 apparently do not impact nuclear YAP retention in alternate mesothelioma cell line models (38) and the osteosarcoma U2OS cell line (53), respectively, we identified that most of the selective TEAD inhibitors tested did modulate YAP localisation 24 h post-treatment. This effect is observed by decreased nuclear/cytoplasmic YAP ratio

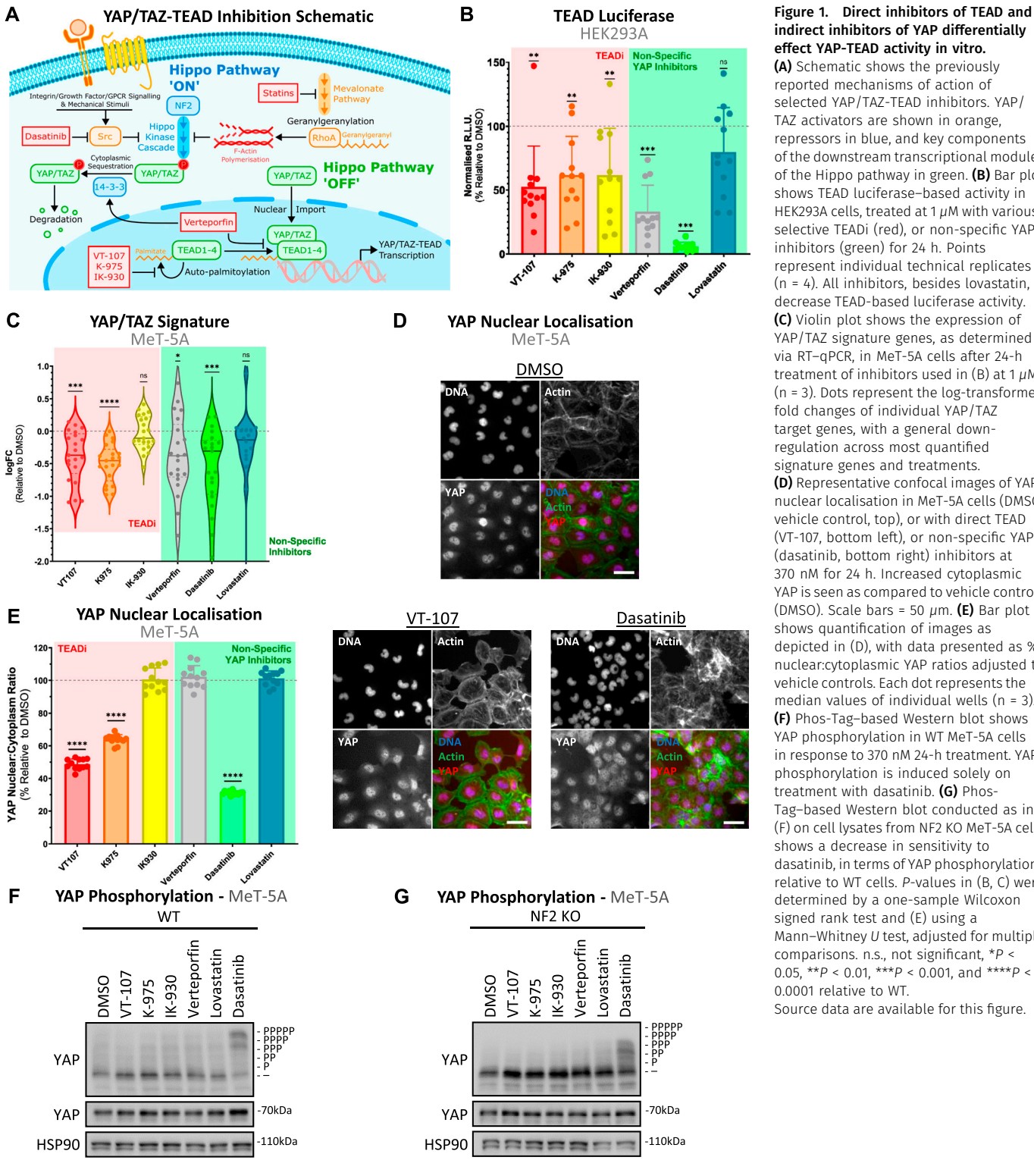

**Figure 1. Direct inhibitors of TEAD and indirect inhibitors of YAP differentially effect YAP-TEAD activity in vitro.**
**(A)** Schematic shows the previously reported mechanisms of action of selected YAP/TAZ-TEAD inhibitors. YAP/TAZ activators are shown in orange, repressors in blue, and key components of the downstream transcriptional module of the Hippo pathway in green. **(B)** Bar plot shows TEAD luciferase–based activity in HEK293A cells, treated at 1 $\mu$M with various selective TEADi (red), or non-specific YAP inhibitors (green) for 24 h. Points represent individual technical replicates (n = 4). All inhibitors, besides lovastatin, decrease TEAD-based luciferase activity. **(C)** Violin plot shows the expression of YAP/TAZ signature genes, as determined via RT–qPCR, in MeT-5A cells after 24-h treatment of inhibitors used in (B) at 1 $\mu$M (n = 3). Dots represent the log-transformed fold changes of individual YAP/TAZ target genes, with a general down-regulation across most quantified signature genes and treatments. **(D)** Representative confocal images of YAP nuclear localisation in MeT-5A cells (DMSO, vehicle control, top), or with direct TEAD (VT-107, bottom left), or non-specific YAP (dasatinib, bottom right) inhibitors at 370 nM for 24 h. Increased cytoplasmic YAP is seen as compared to vehicle control (DMSO). Scale bars = 50 $\mu$m. **(E)** Bar plot shows quantification of images as depicted in (D), with data presented as % nuclear:cytoplasmic YAP ratios adjusted to vehicle controls. Each dot represents the median values of individual wells (n = 3). **(F)** Phos-Tag–based Western blot shows YAP phosphorylation in WT MeT-5A cells in response to 370 nM 24-h treatment. YAP phosphorylation is induced solely on treatment with dasatinib. **(G)** Phos-Tag–based Western blot conducted as in (F) on cell lysates from NF2 KO MeT-5A cells shows a decrease in sensitivity to dasatinib, in terms of YAP phosphorylation relative to WT cells. *P*-values in (B, C) were determined by a one-sample Wilcoxon signed rank test and (E) using a Mann–Whitney *U* test, adjusted for multiple comparisons. n.s., not significant, \*$P < 0.05$, \*\*$P < 0.01$, \*\*\*$P < 0.001$, and \*\*\*\*$P < 0.0001$ relative to WT.
Source data are available for this figure.

(Fig 1D), indicating a reduction in the level of transcriptionally active YAP, with the maximal response generally observed at the lowest concentration point included (41.2 nM; Fig S1B). Reduction in

YAP nuclear localisation (Fig 1E) broadly matches decreased YAP/TAZ signature gene expression (Fig 1C), with a clear effect observed when cells were treated with pan-TEAD inhibitors. However, in

contrast to verteporfin's effect on YAP/TAZ-TEAD gene expression, there was no significant decrease in nuclear YAP, with dasatinib being the sole non-selective YAP/TAZ-TEAD inhibitor inducing YAP cytoplasmic retention in MeT-5A cells (Fig 1E).

With the inhibitory potential of these compounds established within MeT-5A cells, we next assessed this effect within the clinically relevant context of *NF2* loss. Notably, loss of NF2 had little impact on sensitivity to YAP/TAZ-TEAD inhibition in terms of the expression of YAP/TAZ signature genes, with just a decrease in sensitivity to K-975 conserved in both clones (Fig S1C), whereas NF2 KO cells exhibited slightly enhanced sensitivity to selective TEAD inhibition on YAP nuclear localisation (Fig S1D). Though this effect was significant, it is modest, whereas sensitivity to dasatinib is conversely noticeably decreased in NF2 KO cells.

YAP's principal nuclear/cytoplasmic regulation primarily takes place via LATS1/2-mediated inhibitive phosphorylation on five serine residues (50, 54). To interrogate the manner with which YAP inhibition is mediated, we analysed cells after 24-h treatment at both 370 nM and 1 $\mu$M. Lysates were analysed by Phos-Tag Western blotting, a sensitive technique that allows for the identification of the phosphorylation state of YAP (50, 54). These data show that dasatinib is the sole tested inhibitor that induces YAP phosphorylation at low concentrations (Fig 1F), whereas lovastatin treatment results in YAP phosphorylation only at higher concentrations (Fig S1E). This YAP phosphorylation indicates that dasatinib's potent inhibitory effect on YAP/TAZ-TEAD (Fig 1F), as well as lovastatin's more modest effect (Fig S1E), may be mediated via the core kinase module of the Hippo pathway. To explore this, we conducted Phos-Tag–based Western blots using lysates from similarly treated LATS1/2 double-knockout (dKO) HEK293A cells (55, 56, 57). Cells lacking LATS1/2 have, as expected, less phosphorylated YAP, indicating nuclear (55, 56, 57) and therefore hyperactive YAP (Fig S1F). These cells also show no YAP phosphorylation response to treatment with lovastatin or dasatinib, whereas cells with intact LATS1/2 exhibit TEAD-independent phosphorylation of YAP on treatment (Fig S1F). Combined, the results highlight that YAP phosphorylation induced by lovastatin or dasatinib treatment is dependent on LATS1/2 kinases and therefore likely mediated via the Hippo pathway kinase cascade.

Interestingly, the increase in phosphorylated YAP observed in NF2 KO MeT-5A cells upon dasatinib treatment is diminished relative to WT parental MeT-5A cells (Fig 1F and G). This blunted response is consistent with the decreased response of NF2 KO cells to dasatinib treatment in terms of YAP nuclear localisation (Fig S1D). These findings suggest that NF2 loss may result in resistance to inhibition of YAP/TAZ-TEAD via modulators of the Hippo pathway kinase module. In contrast, our data highlight that NF2-deficient cells are similarly vulnerable to TEAD inhibition, as compared to those with intact NF2.

### YAP/TAZ-TEAD inhibition disrupts the tumorigenic potential of NF2-deficient mesothelial cells

With the activity of the candidate compounds established, we next sought to determine the in vitro anticancer potential of candidate therapeutics in the context of NF2-deficient mesothelioma. Initially, this was assessed by quantifying YAP/TAZ-TEAD inhibition's impact on cellular viability. Proliferation assays performed over the course of 72 h (Fig 2A) reveal that selective TEAD inhibitors generally have little effect on cell growth (Fig 2B). VT-107 and to a lesser extent K-975 both show a consistent, though modest, reduction over the course of 72-h treatment, primarily at concentrations approaching 10 $\mu$M. This is in stark contrast to the non-selective YAP inhibitors, which cause a complete cessation of growth at concentrations >1 $\mu$M (Fig 2B).

On NF2 loss, sensitivities across all treatments remain almost equivalent to WT MeT-5A cells (Figs 2C and S2A). This is somewhat unexpected, as recent findings have indicated that NF2-deficient mesothelioma cells are more sensitive to TEAD inhibitors (58). Importantly, our studies are conducted on an isogenic background and are therefore not impacted by the complexities of comparing different heterogeneous cell lines with varying epigenetic backgrounds and underlying genetic diversity beyond *NF2* status. Combined, these additional factors might modulate sensitivity to TEAD inhibition, obfuscating nuanced molecular dynamics. Interestingly, loss of viability induced by non-selective YAP inhibitors appears to be YAP-independent, with YAP KO MeT-5A cells exhibiting similar sensitivities as WT cells to compounds tested (Fig 2D). YAP independence therefore might explain the initial observation that NF2 KO cells are equally sensitive to dasatinib in terms of proliferation (Figs 2C and S2A), despite the decrease in sensitivity observed in NF2-deficient relative to NF2-intact cells in the context of YAP activation (Figs 1F and G and S1D).

Given NF2's role in regulating YAP/TAZ in response to mechanical stimuli, including contact inhibition, substrate stiffness, and the mechanotransduction mediated by the YAP/TAZ-TEAD axis (6), we next focused on the establishment of a functional assay capable of quantifying cellular response to these cancer-adjacent mechanical stresses. Previously, we identified NF2 as an upstream mechanoregulator in mesothelial cells, and that mesothelial NF2 loss is sufficient to drive YAP-mediated anchorage-independent growth (6). As the soft agar assay is challenging to scale up for high-throughput assays, we sought to establish an analogue approach. We therefore implemented a spheroid formation assay to analyse the effects of the selected inhibitors in a scalable manner. When MeT-5A cells are cultured in an ultra-low attachment setting, they readily form spheroids, with spheroids exhibiting less nuclear YAP than cells cultured in 2D (Fig S2B). Interestingly in spheroids, NF2 loss leads to relative enhanced nuclear YAP signal (Fig S2C), suggesting a resistance to the reduction in nuclear YAP observed in WT spheroids. To assess how these differences in YAP activity may drive distinct spheroid phenotypes, we compared the growth of spheroids with intact NF2 and YAP to those with either KO of NF2 or YAP (Fig 2E). In addition to a relative increase of nuclear YAP in MeT-5A spheroids on NF2 loss, there is a concurrent increase in spheroid size (Fig 2F). In contrast, YAP-deficient spheroids exhibit a clear decrease in size (Fig 2F). The increased growth observed in NF2-deficient spheroids recapitulates key aspects of the 3D tumour environment and therefore reflects tumorigenic potential (59), possibly mediated via enhanced YAP activity.

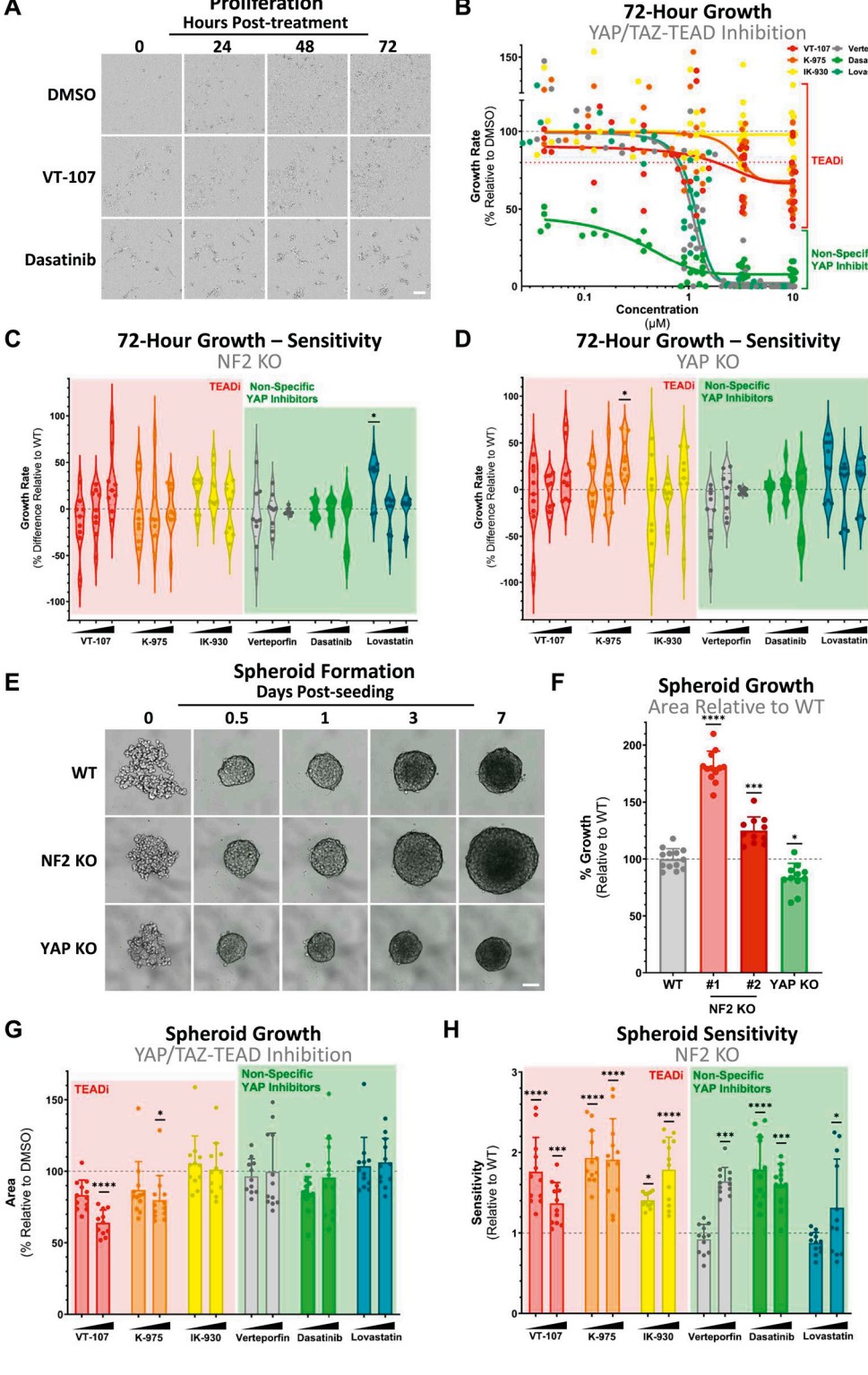

**Figure 2. Specific TEAD-selective inhibition decreases NF2-deficient cancer-relevant phenotypes.**
**(A)** Representative images of 2D proliferation in WT MeT-5A cells at 10 $\mu$M YAP-TEAD inhibition. Scale bar = 50 $\mu$m. **(B)** Dose–response curve of data from images as shown in (A) reveals dose-dependent inhibition of growth selectively in all non-specific YAP inhibitors (green) tested. Direct TEADi (red) has a modest impact on cell growth in 2D compared with non-specific YAP inhibitors (green). Points represent the growth curve statistics of individual wells (n = 3). **(C)** Violin plot shows cell proliferative sensitivity of NF2 KO MeT-5A cells, with individual points representing the percentage relative decrease in cell growth of individual wells (n = 3) in comparison with those in WT cells (B). Negative percentage values indicate an increase in growth rates on treatment, and hence reduced sensitivity, relative to WT. Little change in sensitivity is observed upon loss of NF2. **(D)** Violin plot as in (C) showing sensitivity to YAP/TAZ-TEAD inhibition in YAP KO relative to WT MeT-5A cells, in terms of cell growth rates in 2D. **(E)** Representative images of spheroid formation in WT (top), NF2 KO (middle), and YAP KO (lower). Scale bar = 50 $\mu$m. **(F)** Quantification of spheroid area, as imaged in (E), shown as bar plots with individual points representing the area of individual spheroids at day 7 adjusted to size at 24 h, normalised to the mean area of WT spheroids (n = 3), in WT (grey), two NF2 KO clones (red), and YAP KO (green) MeT-5A cells. Genotypes form spheroids of varying sizes, with spheroids containing NF2 loss (larger) and YAP loss (smaller), respectively. #1 and #2 NF2 KO are independently generated clones. **(G)** Bar plot shows sensitivity of WT MeT-5A spheroids to YAP-TEAD inhibition. Each dot represents the size of an individual spheroid at day 7, relative to size at 24 h to account for variable seeding (n = 3). Spheroid size is modestly decreased upon high-concentration TEAD inhibition. **(H)** Bar plot as in (G) shows sensitivity of NF2 KO spheroids to YAP-TEAD inhibition relative to WT. NF2 KO MeT-5A spheroids have enhanced sensitivity to most tested inhibitors. Inhibitors tested in (B) were used from 41.2 nM to 10 $\mu$M via 1:3 serial dilution, in (C, D) at 1.1, 3.3, and 10 $\mu$M and in (G, H) at 1 and 10 $\mu$M and are compared with vehicle control (DMSO). $P$-values in (B, C, D, F, G, H) were determined by a Mann–Whitney $U$ test, adjusted for multiple comparisons. n.s., not significant, $*P < 0.05$, $**P < 0.01$, $***P < 0.001$, and $****P < 0.0001$ relative to WT. Source data are available for this figure.

Next, having established a quantifiable cancer-relevant phenotype robustly driven by NF2 loss, we tested spheroid growth in the presence of YAP/TAZ-TEAD inhibitors to quantify the impact of inhibition on the tumorigenic capacity of spheroids. WT MeT-5A cells exhibit partial sensitivity selectively to pan-TEAD inhibitors (Fig 2G), with a significant reduction in spheroid growth observed

on treatment with VT-107 and K-975 at high concentrations (10 $\mu M$). Interestingly, NF2 KO MeT-5A cells were more sensitive to nearly all treatments (Figs 2H and S2D). Beyond sensitivity, our data highlight that NF2 KO cells exhibit a marked decrease in spheroid area, with an enhanced effect observed with on-target TEAD inhibition as compared to indirect YAP/TAZ-TEAD inhibitors (Fig S2E). This inhibitory effect is sufficient to revert NF2-deficient spheroids to sizes equivalent to YAP KO spheroids.

These findings collectively point to the spheroid formation assay as effective to evaluate and probe YAP/TAZ-TEAD-driven tumorigenic capacity.

### Determining the role of YAP and TEAD isoforms on YAP/TAZ-TEAD inhibitor efficacy

Given that NF2-mutant in vitro anchorage-independent growth is dependent on YAP (6, 52, 60), we next sought to establish the extent to which YAP/TAZ-TEAD influence the observed NF2-deficient phenotype. To confirm YAP's role in driving NF2 KO spheroid formation, we exogenously expressed constitutively active YAP (YAP-5SA) in MeT-5A WT cells. YAP-5SA contains point mutations across five LATS1/2 target serine residues, rendering YAP non-phosphorylatable by LATS, and therefore hyperactive (52). The expression of YAP-5SA in MeT-5A cells (Fig 3A) reveals that hyperactive YAP drives spheroid growth, closely phenocopying NF2 KO spheroids (Fig 3B). To resolve whether this is driven by TEAD activity targetable via therapeutic inhibition, YAP-5SA spheroids were treated with selective TEAD inhibitors. Spheroids with hyperactive YAP phenocopy the response to drug inhibition observed upon NF2 loss, with enhanced sensitivity to TEAD inhibition relative to WT cells (Figs 2H and 3C).

We next assessed spheroid growth in NF2/YAP double-knockout (dKO) cells (Fig 3D and E). When YAP is concurrently lost in NF2 KO cells, a reversal of the NF2 KO phenotype is observed, with NF2/YAP dKO spheroids exhibiting a dramatic decrease in spheroid growth relative to NF2 KO (Fig 3F). As such, NF2/YAP dKO spheroid sizes were restored roughly to those observed in YAP KO spheroids. Taken together, these observations further reinforce that YAP is the primary effector of NF2 KO spheroid growth and, given the enhanced sensitivity to TEAD inhibition observed in the presence of hyperactive YAP, may be targetable via inhibiting YAP-TEAD transcription.

The TEAD family consists of four proteins in vertebrates, in humans termed TEAD1-4 (13). Recent early-stage clinical evaluation highlights that pan-TEAD and TEAD isoform specificity might be a defining clinical feature in toxicity and efficacy (61, 62). The implications of isoform-specific or pan-TEAD targeting in cancer are therefore of importance, with pan-TEAD likely being more effective but potentially exposing patients to a higher risk of toxicity, particularly to renal damage associated with TEAD inhibitor treatment (61). Assessing TEAD isoform functional importance and drug specificity is therefore critical during drug development to guide optimal clinical responses. These implications prompted us to generate NF2/TEAD isoform-specific dKO cells, with a focus on TEAD isoforms that have readily available antibodies and are robustly expressed and detected in MeT-5A cells. TEAD1 and TEAD4 were specifically recognised and targeted

(Fig 3D), with both TEAD1 and TEAD4 implicated in the progression and development of various cancers (63, 64, 65). Notably, TEAD4 loss leads to a consistent increase in protein levels of TEAD1 (Fig 3D), with quantification of protein levels highlighting the strength of this compensatory effect, while revealing a less consistent up-regulation of TEAD4 on TEAD1 loss and down-regulation of both TEAD1 and TEAD4 on YAP loss (Fig 3D and E). This finding has ramifications for mesothelioma, and potential general cancer therapy, given the possibility of compensatory up-regulation to counter the inhibition of single components within the pathway.

In contrast to the dramatic YAP loss phenotype, loss of TEAD1 or TEAD4 leads to a more modest decrease in spheroid growth (Fig 3G), reverting spheroid size closer to that of WT spheroids. This reduction in growth is particularly pronounced in TEAD4 KO cells. When compared to NF2/YAP dKO, these results suggest complete and partial dependence of NF2-deficient spheroid overgrowth on YAP and individual TEAD isoforms, respectively. We in addition assessed the growth of various KO and dKO MeT-5A cells in 2D, revealing that growth on tissue culture–treated plastic appears largely independent of YAP-TAZ/TEAD activity. The overall combinatorial loss of NF2 and YAP/TEAD isoforms appears insufficient to affect growth rates, with only a single NF2/TEAD4 dKO clone exhibiting a significantly affected growth rate (Fig S3A). These results confirm that the observed spheroid growth defects are not dependent on base-level genotype differences, consistent with the 2D YAP-independent growth rates observed when testing YAP/TAZ-TEAD inhibitors (Fig 2B–D).

To assess which of the four TEAD isoforms, if any, coordinate the hypersensitivity of NF2-deficient spheroids to TEAD inhibition, we treated NF2/TEAD1 and NF2/TEAD4 dKO MeT-5A spheroids with TEAD inhibitors and compared response to parental genotypes. If the observed response to TEAD inhibition is mediated via YAP-TEADs, a limited additive effect of spheroid reduction combining TEAD inhibition with KO of critical TEAD isoforms would be expected. This diminished additive effect may be clearly observed qualitatively, where vehicle control–treated NF2/TEAD dKO genotypes have achieved a near-maximal reduction in spheroid area relative to parental NF2 KO, with further TEAD inhibition resulting in only a modest decrease (Fig S3B). Although TEAD4 is more critical for the establishment of the observed NF2-deficient phenotype, with a more complete reversal on concurrent TEAD4 loss as compared to TEAD1 (Fig 3G), quantifying dKO sensitivities reveals that the loss of either isoform is sufficient to offset the increased sensitivity of NF2 KO spheroids to selective TEAD inhibition (Figs 3H and I and S3C and D). More specifically, loss of TEAD1 leads to a blunting of NF2 KO spheroid sensitivity to selective TEAD inhibition, with all compounds exhibiting reduced sensitivity at one or both concentrations tested (Figs 3H and S3C). Noteworthy, this effect is also observed upon TEAD4 loss (Figs 3I and S3D), even with IK-930, purported to be a selective TEAD1 inhibitor. The decrease in sensitivity observed in NF2/TEAD4 dKO spheroids suggests that TEAD4 may coordinate, at least in part, some IK-930 response (Fig 3I).

In summary, our data further validate the role of YAP and TEADs in coordinating the tumorigenic capacity of NF2 loss in vitro, with YAP in particular indispensable for the enhanced spheroid-forming capability of NF2-deficient cells. Though there is likely

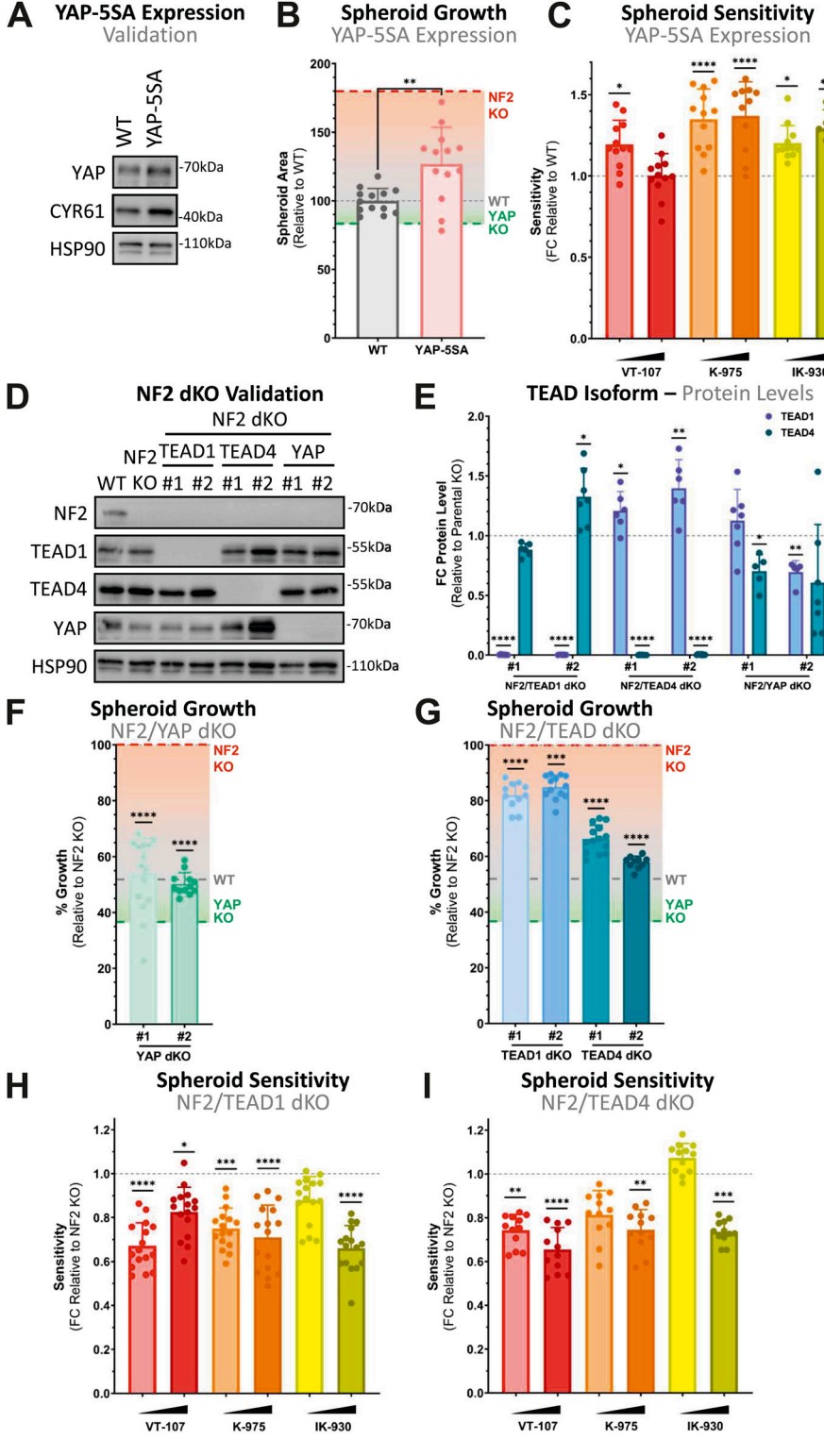

**Figure 3. YAP and TEAD differentially orchestrate NF2-deficient spheroid sensitivity.**

**(A)** Western blots from MeT-5A cells stably expressing YAP-5SA, a hyperactive form of YAP. An increase in protein levels of YAP and CYR61, a downstream transcriptional target of YAP-TEAD, confirms induction of YAP-5SA expression. **(B)** Bar plot shows the relative spheroid growth of YAP-5SA expressing MeT-5A spheroids, compared with the WT control. Each dot represents the area of an individual spheroid at day 7, adjusted to size at 24 h and normalised to mean WT spheroid area (n = 3), whereas gradients show % growth relative to parental NF2 single KO (red), WT (grey), and YAP KO (green). **(C)** Bar plot shows sensitivity of MeT-5A spheroids expressing YAP-5SA to TEAD inhibition. Points represent the sensitivity (decrease in growth on treatment) of individual spheroids, relative to WT control spheroids (n = 3). **(D)** Western blots from gene edited combinatorial KO clones show KO of YAP and TEAD isoforms alongside NF2 in MeT-5A cells. **(E)** Bar plot shows the quantification of TEAD isoform protein levels across Western blots as in (D), with each point representing an individual biological replicate (n = 7). **(F, G)** Bar plots, as in (B), show the quantification of relative spheroid growth comparing the impact of YAP (F) or TEAD (G) KO in NF2-deficient MeT-5A spheroids, with individual spheroid areas normalised to parental NF2 KO. Gradients as in (B) show % growth relative to single KO genotypes. **(H, I)** Bar plots, as in (C), show sensitivity of NF2/TEAD1 dKO (H) and NF2/TEAD4 dKO (I) MeT-5A spheroids to TEAD inhibition, with each dot representing an individual spheroid (n = 3). Loss of either TEAD isoforms results in a decrease in sensitivity to TEAD inhibition. Spheroids in (C, H, I) were treated for 24 h with 1 and 10 μM of selective TEAD inhibitors. P-values in (B, C, F, G, H, I) were determined by Dunnett's multiple comparison test and in (E) using a one-sample t test with a hypothetical mean of 1. n.s. = not significant, *P < 0.05, **P < 0.01, ***P < 0.001, and ****P < 0.0001. Source data are available for this figure.

redundancy among the TEAD isoforms, the enhanced response to TEAD inhibition in NF2-deficient spheroids is blunted when TEAD isoforms are concurrently eliminated. The additional observation that a clear and significant down-regulation of YAP/TAZ signature

genes is observed in NF2/TEAD dKO spheroids relative to parental WT spheroids (Fig S3E) further reinforces the role of TEAD in spheroid formation. Collectively, we show that YAP/TAZ-TEAD activity is critical in driving the aberrant spheroid growth observed

in NF2-deficient cells, which represent a clinical subgroup of PM patients, and that the high-content MeT-5A–based spheroid formation assay is ideal to quantify on-target YAP/TAZ-TEAD inhibition.

### Morphometric profiling predicts selectivity of YAP/TAZ-TEAD inhibitors

Off-target effects are a major issue for drug development, directly impacting multiple levels of clinical transition, with common issues including toxicity, specificity, and efficacy affecting progress (66, 67, 68, 69). Inhibitor treatments that exhibit broad off-target effects often induce morphological changes beyond their effect on proliferation, stemness, and therapeutic resistance. This is exemplified in the disruption of actin organisation in MeT-5A cells on treatment with dasatinib (Fig 1D), consistent with previous observations across multiple cell lines (31, 70). In addition, a highly disrupted morphology is observed in spheroids on treatment with high concentrations of non-selective inhibitors (Fig S4A), highlighting a recurrent phenotype associated with off-target effects unrelated to YAP/TAZ-TEAD inhibition.

As shifts in cellular morphology are predictive of broad pharmacological effects (71) and can be leveraged to infer detailed molecular effects of treatment, we next sought to quantify the morphological disruption on YAP/TAZ-TEAD inhibition in our cellular model system. To initially confirm the phenotype observed in MeT-5A spheroids (Fig S4A), we performed a quantitative analysis of the morphological shift of spheroids on treatment with YAP/TAZ-TEAD inhibitors. Brightfield images were segmented, and morphometric characteristics were extracted and processed according to Joint Undertaking in Morphological Profiling (JUMP) pipelines (72 Preprint). This revealed a distinct morphological clustering within non-specific YAP inhibitors, whereas selective TEAD inhibition results in a phenotype that readily clusters with vehicle control cells (Fig S4B).

Cell Painting is a newly developed approach to morphological profiling, which has been incorporated into high-throughput imaging assay pipelines to impute mechanisms of action (73), bioactivity (74), and toxicity (75, 76) in recent drug discovery efforts. To quantify the morphological perturbations induced on inhibition of YAP/TAZ-TEAD in greater resolution, we undertook Cell Painting to generate high-dimensionality datasets. Cell Painting involves the staining of cells with six dyes (Fig S4C) and was performed in accordance with protocols established by the Joint Undertaking in Morphological Profiling Cell Painting (JUMP-CP) consortium (77). Principal component analysis (PCA) of high-dimensional morphometric data reveals a morphological shift on treatment with non-selective inhibitors of YAP (Fig 4A). To quantify this shift, the correlation distance across all morphological features was computed between treated wells. The resulting correlation distance matrix shows a clustering of non-selective and high-concentration inhibitors, which share morphological similarities (Fig 4B). Interestingly, there was little clustering in vehicle control and low-concentration selective TEAD inhibitor wells. This suggests that morphological uniformity may be a metric of non-specific inhibition of YAP/TAZ-TEAD. Quantification of uniformity, as defined by the mean correlation

distance between like-treated cells, reveals significantly enhanced morphological uniformity on high-concentration, as well as low-concentration dasatinib and lovastatin, treatment (Fig 4C). Similar uniformity profiles were observed in NF2 KO cells on treatment (Fig S4D). To assess whether this observed uniformity is YAP-dependent or YAP-independent, we quantified morphological uniformity on treatment in YAP KO MeT-5A cells (Fig 4D). This reveals similar trends, with both dasatinib and lovastatin inducing a shift to more morphologically similar cells, thus indicating YAP independence. Strikingly, high-concentration K-975 and verteporfin treatment induced no such effect in YAP-deficient cells in contrast to WT, which may suggest a degree of morphological disruption mediated via YAP-TEAD inhibition.

With the broad impact of treatment on cell morphology established (Fig 4A–D), we next performed in-depth analysis of specific features disrupted on inhibition of YAP/TAZ-TEAD. For this, we implemented a modified robust Z′ scoring, a parameter typically used in quality control of drug screening, to extract features for each treatment that resolve distinctly from untreated controls. The Z′ scoring reveals a disruption of a high percentage of features on treatment with non-specific YAP and high concentrations of TEAD-specific inhibitors (Fig 4E), consistent with broad profile analysis. Interestingly, the percentage of disrupted features is generally consistent when YAP is lost, particularly in dasatinib-treated cells (Fig 4E), suggesting that this morphological disruption is YAP-independent. To quantify this YAP specificity in detail, we adjusted modified Z′ statistics in WT to YAP KO cells. This revealed a range of conserved features that were consistently disrupted in a YAP-independent manner specifically on treatment with the non-specific YAP inhibitors dasatinib and lovastatin (Fig 4F). Conversely, a smaller proportion of features were YAP-dependent, which were disrupted on treatment with both selective TEAD inhibitors, with minimal overlap between the various TEAD inhibitors, and non-specific YAP inhibitors. Intriguingly, this suggests that different sets of features may be used to quantify YAP-dependent and YAP-independent morphological changes, with verteporfin and both selective TEAD inhibitors showing less off-target YAP effects than dasatinib and lovastatin.

## Discussion

Given the critical role the Hippo pathway and, more specifically, its downstream effectors YAP/TAZ-TEAD play in the progression of mesothelioma and other cancers, there is a clear need to develop pipelines capable of assessing the efficacy and specificity of newly developed inhibitors to streamline positioning to clinical use. The realisation of the complex multiparametric regulations (36, 44, 78, 79) and feedback loops (80) within the Hippo pathway, including multiple regulators of protein turnover of key components (81, 82, 83), further emphasises the necessity to investigate these compounds' cellular effects in detail. Although in vitro studies assessing the efficacy of TEAD inhibition in NF2-deficient mesothelioma cells have been described (58), these are limited by the heterogeneous genomic and transcriptomic landscapes associated with cell lines of

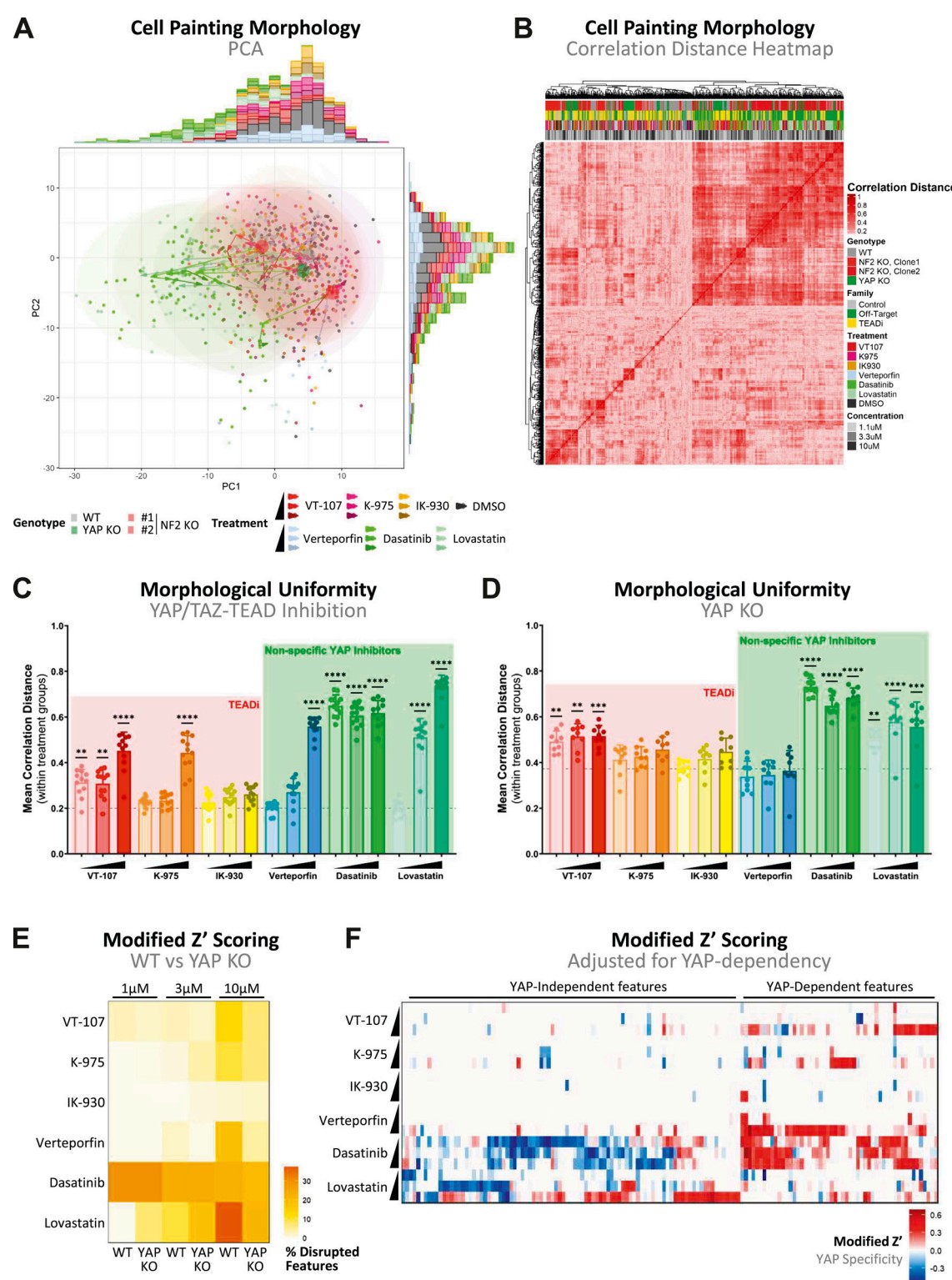

**Figure 4. Morphological disruption predicts specificity of YAP/TAZ-TEAD inhibitors.**
**(A)** PCA plot shows morphological profiles of MeT-5A cells cultured in 2D, with features collapsed via dimensionality reduction to allow visualisation. Each point represents a single well (n = 3), with highlighted circles representing the centre of DMSO-treated cells coloured according to genotype. Ellipses show the spread of treatments irrespective of genotype, and arrows show trajectories across increasing concentrations of treatment for each genotype. **(B)** Heatmap shows correlation distance of computed features between wells from morphometric data represented in (A). Clustering of cells, each representing a single well (n = 3), was performed via complete linkage, with annotations (top) showing the genotype, inhibitor family, name of treatment, and treatment concentration for each well. **(C)** Bar plot portraying morphological uniformity of WT MeT-5A cells on YAP/TAZ-TEAD inhibition. Each point represents a single well (n = 3), with uniformity calculated as the mean correlation

mixed origins. This heterogeneity potentially obfuscates critical, nuanced molecular dynamics of inhibiting the transcriptional machinery of the Hippo pathway. Here, we present a pipeline within an isogenic model of loss of upstream and downstream components within the Hippo pathway, allowing the detailed interrogation of the molecular interplay of the individual constituents that comprise this tumorigenic pathway. The observations made here consolidate that 2D cellular models may be insufficient to support evaluation of YAP/TAZ-TEAD inhibitors, particularly in contexts where mechanosensory conditions do not reflect real-life tissues, thus highlighting the need for more complex model systems. Our data also highlight the clear distinctive changes in YAP/TAZ dynamics between 2D and 3D cultures (Figs 1D and E and S2B). We in addition describe functional assays for quantifying the selective YAP/TAZ-TEAD inhibition of tumorigenic capacity, which have shed light into the specifics of efficacy and selectivity of the core compound library used to target the major players within the pathway.

Prior multiomic and genetic analyses have highlighted the in vivo complexities of pleural mesothelioma (6, 7, 8, 9, 84, 85). The unfortunate continued lack of curative therapies for this deadly disease is likely due to late-stage diagnosis and the sparsity of complementary preclinical models representing the disease required for the predictive evaluation of drug targets before clinical evaluation in patients. Notably, our observations predict that targeting the transcriptional complex directly via inhibition of TEAD, including via its autopalmitoylation site, is more specific, and thereby likely more effective, than targeting the upstream kinase module. Our observations also correlate well with clinical data, including the lack of activity and overall unwelcome association with pulmonary toxicities with dasatinib treatment in unselected pleural mesothelioma patients (86). Our platform combines an extensive, in vitro–based isogenic cellular model allowing for specificity evaluation, cancer-relevant assays where tumour suppressors are functionally active, high-content imaging, and Cell Painting. Collectively, these facilitate direct evaluation of YAP/TAZ-TEAD(1-4) targeting drugs and are feasible to implement alongside synergistic combinatorial drug evaluations. We are hopeful that our approach provides a milestone within the development of mesothelioma treatments, as we appear to potentially evaluate potency of inhibition (87) and clinical efficiency. The development of the platform thereby might inform future clinical trials and overall complement ongoing efforts to develop advanced cellular models (88, 89) and ultimately curative therapies.

Results point to K-975 as an exemplar inhibitor of TEAD autopalmitoylation, with VT-107 exhibiting equivalent potency alongside off-target effects at higher concentrations. Both inhibitors, however, show some lack of specificity at high concentrations (Fig 4B and F), leading to a partial breakdown between dose and response. This is evident in the diminished YAP cytoplasmic sequestration response to K-975 treatment at concentrations above 1 $\mu$M (Fig S1B). A similar phenomenon is also observed in the heightened sensitivity to VT-107 treatment in hyperactive YAP spheroids at 1 $\mu$M, which is diminished at 10 $\mu$M (Figs 2H and 3C). Despite this, YAP-dependent effects are enhanced in a dose-dependent manner (Fig 4F), indicating a need for balancing dosage to maximise therapeutic efficacy while avoiding issues with tolerance in clinical use for PM patients.

Conversely, although IK-930 treatment is not associated with off-target effects and therefore a favourable safety profile, its potency is markedly less relative to alternative TEAD inhibitors (Fig 1B–E). Despite previously reported selectivity for TEAD1 (90), the observation that loss of TEAD4 impacts the sensitivity of spheroids to IK-930 to the same extent of TEAD1 (Fig 3H and I) indicates some degree of inhibition of additional TEAD isoforms. Our findings also highlight the apparent compensatory mechanisms between TEAD isoforms, a critical element to consider for drug development (Fig 3D). This discovery may explain the promiscuity observed in IK-930, suggesting a compensatory up-regulation of, and therefore potential dependence on, TEAD1 on loss of TEAD4. Compensatory mechanisms may also shed some light on isoform specificity and potential for therapeutic targeting, with a more consistent compensatory response, alongside a greater decrease in spheroid area (Fig 3G), on loss of TEAD4 as compared to TEAD1. In contrast, there are a less clear compensatory up-regulation of TEAD4 and diminished impact on spheroid growth on loss of TEAD1, indicating that the absence of TEAD1 may be more well tolerated in NF2-deficient mesothelial cells. Interestingly, our results echo preliminary results from clinical studies of IK-930, which indicate that, despite being well tolerated, IK-930 has not proved effective in patients enrolled in clinical trials (91). Collectively, these results suggest the likely clinical potential of TEAD4-specific therapeutic inhibitors within the context of mesothelioma, whose potency may be enhanced relative to less effective TEAD1-specific inhibitors and whose specificity may be beneficial to avoid the kidney toxicity associated with pan-TEAD inhibition (61). Currently, most of the TEAD inhibitors exhibit pan-TEAD efficacy, and there is therefore a further need for the development of isoform-specific, or combinatorial, therapeutics for preclinical testing and subsequent clinical deployment. Within our cellular model, loss of no single TEAD isoform is sufficient to reverse the NF2-deficient oncogenic phenotype to the same degree as YAP loss (Fig 3F and G). Our data also highlight the decrease of TEAD protein levels in YAP-deficient mesothelioma cells (Fig 3E), suggesting the absence of an equivalent compensatory mechanism and hence less likely resistance development if direct targeting of YAP were pharmacologically feasible.

distance of a single well to all like-treated wells. **(D)** Bar plot as in (C), showing morphological uniformity on YAP/TAZ-TEAD inhibition in YAP KO MeT-5A cells. **(E)** Heatmap shows the percentage of disrupted features, as defined by modified Z' scores > 0, in WT versus YAP KO MeT-5A cells in response to various YAP/TAZ-TEAD inhibitors. As treatment concentration increases, a greater percentage of features are disrupted across most inhibitors. **(F)** Heatmap shows modified Z' scores of features with known discriminatory potential, defined as those with modified Z' scores > 0, adjusted for YAP dependency. Positive values indicate feature disruption selectively in WT, and not in YAP KO MeT-5A cells, indicating dependence on YAP, whereas negative values show features disrupted to the same or greater degree in YAP KO cells. Cells in (A, C, D, F) were treated for 24 h with 1.1, 3.3, and 10 $\mu$M of compounds. *P*-values in (C, D) were determined by Dunnett's multiple comparison test. n.s., not significant, *$P$ < 0.05, **$P$ < 0.01, ***$P$ < 0.001, and ****$P$ < 0.0001.
Source data are available for this figure.

In comparison with selective TEAD inhibitors, upstream inhibitors of YAP/TAZ-TEAD are associated with sweeping off-target effects, with dasatinib inducing cellular changes well beyond inhibition of the YAP/TAZ-TEAD axis alone. Surprisingly, given the extent of YAP-independent effects previously associated with treatment, verteporfin exhibits a high degree of YAP specificity across a range of functional assays. The convergence between phenotypes in cells treated with verteporfin and selective inhibitors of TEAD reinforces previous findings that verteporfin can act to disrupt YAP-TEAD binding (33), though YAP-independent cytotoxicity is validated within our model system. The observation that high potency inhibitors of TEAD autopalmitoylation also increase cytoplasmic retention of YAP is novel, though consistent with reports that inhibitors of YAP-TEAD interactions concomitantly reduce levels of nuclear YAP in mesothelioma cell lines (92). This phenomenon has been described as likely a result of displacement of YAP from chromatin-resident TEAD, reinforced by the finding that K-975 treatment reduces formation of YAP nuclear condensates, indicative of decreased binding to super-enhancer regions (53).

To conclude, this work provides a framework for quantifying the efficacy of YAP/TAZ-TEAD inhibition for the treatment of mesothelioma. Conventional phenotypic screening approaches, which typically gauge success based on the cytotoxic or cytostatic properties of therapeutics, are impeded by their limited scope and are biased to drugs with broad toxicity profiles (93). As the Hippo pathway integrates a wide range of stimuli (11, 12), it might be particularly prone to these perturbations. The pipeline established here benefits from the incorporation of a confirmed YAP-TEAD–specific phenotypic assay, with results that further validate the advantage of 3D models over conventional 2D systems for interrogating Hippo pathway dynamics and drug development and evaluation (6). The identification of a feature signature of YAP specificity in addition allows for the assessment of potential off-target effects of putative treatments. We expect this pipeline may help expedite the discovery of therapeutics effective, in managing YAP/TAZ-TEAD-driven mesothelioma, including in potential TEAD isoform–specific targeting and in combinatorial synergistic approaches. This approach will likely mitigate some of the many risks of moving into in vivo assessment of inhibitor safety and efficacy.

# Materials and Methods

### Luciferase assay

HEK293A cells were seeded in triplicate in 12-well plates before co-transfection with Gal4-TEAD, 5 × UAS Luciferase reporter, and *Renilla* luciferase constructs, using GenJet transfection reagent (SL100488; SignaGen). Post-transfection, cells were lysed, and luciferase quantification was performed using a Dual-Glo Luciferase Assay kit (E2920; Promega) according to the manufacturer's specifications and measured on a Biotek Synergy HT plate reader, with luminescence adjusted to *Renilla* luciferase.

### Culture maintenance

MeT-5A mesothelial cells and CRISPR/Cas9-mediated KO genotypes were generated and cultured as described in reference 6. HEK293A gene edited cells are described in references 14, 94, 95 and cultured in high-glucose DMEM (21969-035; Gibco) with 2 mM L-glutamine (25030-024; Gibco) and 10% FBS (10500-064; Gibco). Cells were grown in the presence of 100 U/ml of penicillin and 100 $\mu$g/ml of streptomycin (15140-122; Gibco) and incubated at 37°C with 20% $O_2$. Where specified, cells were treated with VT-107 (HY-134957; MedChemExpress), K-975 (HY-138565; MedChemExpress), IK-930, verteporfin (SML0534; Sigma-Aldrich), lovastatin (S2061-SEL; Selleckchem), and dasatinib, gifted by Prof Neil Carragher.

### Gene knockout and ectopic expression of YAP-5SA

Briefly, CRISPR/Cas9-mediated KO was carried out in MeT-5A mesothelial cells as described in reference 6, with the additional use of the following guide RNA (gRNA): 5′-TGGCAGTGG CCGAGACGATC-3′ for *TEAD1* and 5′-CTCAAGGATCTCTTCGAACG-3′ for *TEAD4*. Validation of KO was carried out via Western blotting. The expression of YAP-5SA in MeT-5A cells was performed by lentiviral transduction via pQX system under hygromycin B (30-240-CR; Corning) selection. Lentivirus for transduction was produced in HEK293T cells, harvested 48 and 72 h after transfection using GenJet transfection reagent (SL100488; SignaGen), and filtered using low binding 0.45-$\mu$m SFCA filters (431220; Corning). Selection was carried out via hygromycin treatment 24 h post-transduction to ensure time for the development of resistance.

### Western blotting

Western blots were performed with cell lysates harvested and run using homecast gels, as described in reference 94. PageRuler Prestained Protein Ladder (26616; Thermo Fisher Scientific) was included in Western blots as a scale for protein size. Separated proteins were subsequently transferred from gels to Immobilon-P PVDF membranes (IPVH00010; Millipore) and blocked in 5% milk in TBS-T, with subsequent primary and secondary antibody incubation, washing, and development carried out as described in reference 6. Phos-Tag Western blots were conducted with Phos-Tag reagent (304-93521; Alpha Laboratories) and 10 mM $MnCl_2$ added to each polyacrylamide resolving gel. Primary antibodies used were as follows: YAP (ab52771; Abcam), CYR61 (14479; CST), NF2 (D1D8; CST), TEAD1 (610922; BD Biosciences), TEAD4 (ab58310; Abcam), with HSP90 (BD610418; BD Biosciences) used as a loading control for samples.

### RT–qPCR analysis

Cells were plated and harvested, with cDNA generated as described in reference 6. RT–qPCR assays were carried out using Brilliant III

Ultra-Fast SYBR Green (600883; Agilent) and LightCycler 480 SYBR Green I (04707516001; Roche) RT–qPCR Master Mixes, used according to the manufacturer's directions. IDT primers were custom-designed using templates deposited on PrimerBank ([96]). RT–qPCR was carried out on QuantStudio 5 Real-Time PCR System, with resulting data analysed using R statistical software. Primer sequences were as follows: 5′-GCTCGTTGAGTGAACGGCT-3′ and 5′-CATGAGCTAGTACAACATGAGGG-3′ for *AMOTL2*; 5′-AGTAGAGGAACT GGTCACTGG-3′ and 5′-TGTTTCTCGCTTTTCCACTGTT-3′ for *ANKRD1*; 5′-CCAAGGTGAGCTTTCCCTCG-3′ and 5′-CCTACTAGACCATAGGTCGTCGT-3′ for *ARHGEF17*; 5′-TAGAACAGCCCTTCAGAAAGTGA-3′ and 5′-CGGGGT TGTCTCGACTTAAAAA-3′ for *ASAP1*; 5′-GTGGGCAACCCAGGGAATATC-3′ and 5′-GTACTGTCCCGTGTCGGAAAG-3′ for *AXL*; 5′-CCCTGTGACGAG TCCAAGTG-3′ and 5′-GGTTCCGTAAATCCCGAAGGT-3′ for *CRIM1*; 5′-GAGGCAGAAGTACGGGGTTG-3′ and 5′-CAGGAATCACGGTTTCATGCT-3′ for *DOCK5*; 5′-GGCGCTTCAGGCACTACAA-3′ and 5′-TTGATTGACGGG TTTGGGTTC-3′ for *F3*; 5′-GCTGGTGGACCTAGTACAATGG-3′ and 5′-CTT ACGAGCCGGTCGAAGTTG-3′ for *FJX1*; 5′-AATGCCACTCGCCCTACAC-3′ and 5′-CGTTCTGGTGCAAGTAGCTCT-3′ for *FOXF2*; 5′-GAGAGCAGAAGA CCGAAAGGA-3′ and 5′-CACAACACCACGTTATCGGG-3′ for *GADD45A*; 5′-AGAGCACAGATACCCAGAACT-3′ and 5′-GGTGATTCAGTGTGTCTTCCA TT-3′ for *IGFBP3*; 5′-ACTTTTCCTGCCACGACTTATTC-3′ and 5′-GATGGC TGTTTTAACCCCTCA-3′ for *LATS2*; 5′-TAATTGGCACGGCGACTGTAG-3′ and 5′-GGAGATCAGCTTGTACGGCAG-3′ for *MYOF*; 5′-GCCTGGGAGCTT ACGATTTTG-3′ and 5′-TAGTGCCCTGGTACTGGTCG-3′ for *NT5E*; 5′-CGC CCAAGCCCCTAATGAAG-3′ and 5′-TCCCTCCGTATGTGCATCAGA-3′ for *NUAK2*; 5′-ATGCCTTTTGGTCTGAAGCTC-3′ and 5′-CCCTGTGCTTTCCAC CGAC-3′ for *PTPN14*; 5′-GGGGAACAGTTGAGTAAAACCA-3′ and 5′-ACA ATTTTTCCATACGGTTGGCA-3′ for *RBMS3*; and 5′-CAGCACACTCGATAT GGACCA-3′ and 5′-CCTCGGGCTCAGGATAGTCT-3′ for *TGFB2*. All gene expression values were normalised to *Hypoxanthine Phosphoribosyltransferase 1* (*HPRT1*) expression.

### YAP nuclear localisation high-content image-based assays

Cells were seeded in 384-well μClear plates (781091; Grenier) at variable cell densities determined for each genotype to ensure cells' equivalent confluency over time-points tested. WT MeT-5A cells were seeded at 1,000 cells per well, NF2 KO at 833 cells per well, and YAP KO at 2,000 cells per well. Upon attachment, cells were treated with relevant inhibitor for 24 h before fixing, permeabilising, and staining as described in reference [6]. Fluorescent cells were imaged using the Opera Phenix Plus high-content imaging system (PerkinElmer) at 20x across three biological replicates, with nine fields imaged per well and four wells per sample/condition to act as technical controls. A bespoke CellProfiler ([97]) pipeline was implemented to segment cells and quantify relevant intensities and morphological features. Statistical analyses were conducted using R, with cell-level data trimmed to ensure detectable nuclear and cytoplasmic YAP. Cells were trimmed according to cell contact, with those between 45 and 55% cell–cell contact retained (7.6% cells of 1,381,549 total, with a median of 89 cells/well included for downstream analysis) and variance accounted for post-collation by removing outliers, defined as wells possessing values greater than two median absolute deviations (MAD) from the median of biological replicates. Processed data were plotted with GraphPad Prism.

### Proliferation and spheroid formation assays

For quantification of proliferation, cells were seeded at 2,000, 1,500, or 2,500 cells per well for WT, NF2 KO, and YAP KO MeT-5A cells, respectively, in 96-well μClear plates (655090; Grenier) and treated with relevant inhibitors when attached, 24 h post-seeding. Cells were then imaged over the course of 72 –h, and confluency was calculated using the CELLCYTE X (CYTENA). Growth was quantified by normalising slope statistics from computed logistic growth curves to DMSO-treated wells, with rates of growth <0 adjusted to 0. For spheroid formation assays, cells were seeded at 500 cells per well in ultra-low attachment 96-well plates (CLS7007; Corning) and cultured for 7 d. For live-imaging assays and quantification of spheroid size, cells were incubated and imaged throughout this time period using the CELLCYTE X. To limit the influence of technical artefacts associated with the high variance of this platform, time-points exhibiting spheroid areas greater than two MADs from the median area at that time-point were trimmed, with any well with >50% trimmed time-points defined as outliers and excluded from analysis. 7-d spheroid areas were normalised to 24-h time-points within each well. For treated spheroids, treatment was initiated ~1-h post-seeding, with resulting spheroids imaged at day 7 using EVOS FL Auto 2 Imaging System (Invitrogen) at 10x magnification. Quantification of spheroid size was then achieved using CellProfiler, with the CellPose ([98]) plugin to enhance spheroid segmentation. Brightfield images acquired were then subjected to morphological analysis as described in Cell Painting and morphological analysis, with a pipeline modified for single-channel quantification.

### Cell Painting and morphological analysis

Cell Painting assays were conducted according to established protocols ([77]), with cells seeded as with high-content imaging in 384-well μClear plates (781091; Grenier) and treated with an appropriate inhibitor for 24 h before fixing, staining, and imaging. Image acquisition using the Opera Phenix Plus high-content imaging system (PerkinElmer) was performed with the same setup as described for YAP nuclear localisation high-content image-based assays. Morphometric datasets were then acquired from resulting images using CellProfiler ([97]) pipelines established by JUMP-CP protocols ([72] Preprint, [99]), with the implementation of a modified version of pipelines available at github.com/broadinstitute/imaging-platform-pipelines/tree/master/JUMP_production. The cytominer ([100]) package was then used to trim low variance or highly correlated features (leading to retention of 417 of 3,446 input features; 12.11%) and normalise retained features to vehicle control (DMSO)–treated WT MeT-5A cells. Normalised high-dimensional morphometric datasets were visualised using the circlize ([101]) package in R. Morphological uniformity was computed by calculating correlation distances between all preserved features in each like-treated well. Feature disruption was quantified via a modified robust Z′ scoring, designed to act as a simplified, relative measure of a feature to resolve treated cell morphology from vehicle control (DMSO). This was calculated by removing the scaling constant of 3 of the traditional Z′ score and using median/MAD as opposed to mean/SD as measures across

wells, given no assumption of data to conform to a normal distribution. To implement scoring, the following equation was used:

$$Modifed\,Z' = 1 - \frac{MAD_{Treat} + MAD_{Control}}{Median_{Treat} - Median_{Control}}.$$

Modified $Z'$ scores were adjusted to quantify YAP dependency by subtracting YAP KO from WT scores.

## Data Availability

Raw data, including uncropped WB scans, are included as supplementary materials. Remaining data, including Cell Painting and spheroid morphology datasets, are available upon reasonable request from the corresponding author.

## Supplementary Information

## Acknowledgements

Ongoing research in the Hansen laboratory (CGH) was funded by Worldwide Cancer Research (19-0238) and CSO-LifeArc. This project was initiated by pump prime funding from ISSF3 and JHMRF. MN Fairley and K Purohit were funded by MRC Precision Medicine DTP Studentships. NS Hui was funded by the Martin Lee DTP. S Jia was funded by a scholarship from the Chinese Scholarship Council, and the Edinburgh Global from the University of Edinburgh. MKM's PhD was funded by the Edinburgh Doctoral College Scholarship. We furthermore acknowledge team members for insightful comments and constructive feedback during this study. We acknowledge the technical support and guidance provided by the Institute for Regeneration and Repair (IRR) Flow Cytometry and Cell Sorting Facility staff. Single-cell sorting necessary for clonal selection of expansion of KO cell populations was conducted with support from the QMRI Flow Cytometry and IRR Flow Cytometry and Cell Sorting Facility, University of Edinburgh. For the purpose of open access, the author has applied a Creative Commons Attribution (CC BY) licence to any Author Accepted Manuscript version arising from this submission.

### Author Contributions

R Cunningham: conceptualisation, data curation, formal analysis, validation, investigation, visualisation, methodology, and writing—original draft, review, and editing.

S Jia: data curation, formal analysis, validation, investigation, methodology, and writing—review and editing.

K Purohit: formal analysis and investigation.

MN Fairley: formal analysis and investigation.

MK Maniak: data curation, investigation, and writing—review and editing.

Y Lin: formal analysis and investigation.

NS Hui: data curation, investigation, and writing—review and editing.

RE Graham: methodology.

AG Rossi: supervision and methodology.

J Cholewa-Waclaw: investigation and methodology.

PO Bagnaninchi: resources and supervision.

NO Carragher: resources, methodology, and writing—review and editing.

CG Hansen: conceptualisation, resources, formal analysis, supervision, funding acquisition, investigation, methodology, project administration, and writing—original draft, review, and editing.

### Conflict of Interest Statement

CG Hansen has in the past advised and obtained consultancy fees from Beactica Therapeutics and Ambagon, activities overseen by Edinburgh Innovations. NO Carragher is a former advisor and current shareholder in Amplia Therapeutics Ltd, is a founder, shareholder, and management consultant for Pheno Therapeutics Ltd, and has licenced patents to and holds current research grant funding from Nuvectis Pharma Inc. The companies were not involved and had also no influence in the design of the study. The other authors have nothing relevant to report.

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
