## [Reviewer comments · Life Science Alliance]

Life Science Alliance

Pipeline to evaluate YAP-TEAD inhibitors indicates TEAD inhibition represses NF2-mutant mesothelioma

Richard Cunningham, Siyang JIA, Krishna Purohit, Michaela Fairley, Marcin Maniak, Yue Lin, Ning Hui, Rebecca Graham, Adriano Rossi, Justyna Cholewa-Waclaw, Pierre Bagnaninchi, Neil Carragher, and Carsten Hansen

DOI: <https://doi.org/10.26508/lsa.202503241>

Corresponding author(s): Carsten Hansen, University of Edinburgh

Review Timeline:

Submission Date:	2025-01-30
Editorial Decision:	2025-03-17
Revision Received:	2025-06-11
Editorial Decision:	2025-07-07
Revision Received:	2025-07-12
Accepted:	2025-07-15

Scientific Editor: Tim Fessenden

Transaction Report:

March 17, 2025

Re: Life Science Alliance manuscript #LSA-2025-03241-T

Dr. Carsten G Hansen
University of Edinburgh
Center for Inflammation Research
Institute for Regeneration and Repair
Edinburgh BioQuarter, 4-5 Little France Drive
Edinburg EH16 4UU
United Kingdom

Dear Dr. Hansen,

Thank you for submitting your manuscript entitled "Pipeline to evaluate YAP-TEAD inhibitor potential - TEAD inhibition repress NF2-mutant mesothelioma" to Life Science Alliance. The manuscript was assessed by expert reviewers, whose comments are appended to this letter. We invite you to submit a revised manuscript addressing the Reviewer comments.

Thank you for this interesting contribution to Life Science Alliance. We are looking forward to receiving your revised manuscript.

Sincerely,

B. MANUSCRIPT ORGANIZATION AND FORMATTING:

Reviewer #1 (Comments to the Authors (Required)):

The manuscript describes several cell-based assays to address the effects of inhibitors of the YAP/TAZ pathway. They provide results from 2D and 3D culture models using genetically engineered cell lines for different components of the YAP/TAZ-TEAD pathway. Cell paint technology is used to profile complex morphological phenotypes to distinguish YAP/TAZ-TEAD dependent effects from off-target YAP/TAZ-TEAD-independent effects. From this work the authors conclude that this pipeline of experimental models may help expedite the discovery of therapeutics effective in managing YAP/TAZ-TEAD driven mesothelioma, mitigating some of the risks of moving into in vivo assessment of inhibitor safety and efficacy.

The manuscript presents a large number of different experiments using different experimental models. However, it is difficult to appreciate the significance of the different results. For example, the data from experiments in 2D seem less relevant compared to those in 3D spheroid cultures. The interpretation is also complicated by the absence of dose-response curves. One would like to minimally see a dose-response curve for the observed phenotypes (e.g. Fig 1B). The exception is in figure S1B, but this presents a confusing result for the dose-response effects of the drugs used. In addition, in figure 3B, the dose response relation is very modest if at all present. It seems that IK930 has the most consistent effect, whereas in most other assays, this compound performed less in comparison to VT-107 and K-975.

For many figures difficult to understand the way results presented. In many instances violin plots are used where bar charts would be more informative. It is also not clear what is depicted in these violin plots (replicates?, genes?) and the heterogeneity in the results seems very large and heterogeneous. The choice of drugs included in different experiments is inconsistent. For example, in figure 1C, IK-930 is missing, which is important for comparison with Fig 1B. In figure S1A, Dasatinib not included.

How should we interpret the difference between parental and TEAD knock-out with respect to the Yap/Taz signature genes? The way the data is presented is confusing. For example, when WT is compared to TEAD or NF2 knock-outs, there is no DMSO control.

The effect of perturbation of YAP or TEAD in spheroids with NF2-loss is presented, such experiments are not performed in 2D cell viability experiments.

The study presents perturbation effects of YAP or TEAD in spheroids with NF2 loss but does not include similar experiments in 2D cell viability assays. How does this relate to gene-expression signatures when comparing 2D to 3D (figure 1)? The conclusion that "These findings collectively point to the spheroid formation assay as effective to evaluate and probe YAP/TAZ-TEAD driven tumorigenic capacity" would be stronger if additional data were provided to demonstrate pathway activation/inactivation in 3D cultures.

Figure 2H is unclear in its implications. It should be represented in a manner similar to Figure 2F, where NF2 loss leads to increased growth that is mitigated by TEAD/YAP/TAZ inhibition.

Figures 3B and 3C also suffer from unclear data representation and should be revised for better interpretability.

In addition, in figure 3B, the dose response relation is very modest if at all present. It seems that IK930 has the most consistent effect, whereas in most other assays, this compound performed less in comparison to VT-107 and K-975.

The statement: "Considered together, these findings suggest that NF2-loss may result in a resistance to inhibition of YAP/TAZ-TEAD via activation of the Hippo pathway's core kinase cascade, while leaving NF2-deficient cells similarly vulnerable to TEAD inhibition as compared to those with intact NF2" is difficult to follow. The authors should clarify and refine this conclusion.

The statement on page 8: "Recurrent phenotype associated with off-target effects of YAP/TAZ-TEAD inhibition" appears incorrect. The small molecules used in the study have off-target effects that are independent of YAP/TAZ-TEAD inhibition.

Reviewer #2 (Comments to the Authors (Required)):

This paper uses a standardized isogenic cell system to systematically evaluate hippo pathway inhibitors, particularly those that target TEAD proteins. The strength of the manuscript is in the model system, which eliminates many confounding genetic

variables, and the use of 3D spheroid models, which accentuate the effects of the inhibitors. This is potentially significant, as TEAD inhibitors have entered clinical trials in mesothelioma and other cancers with NF2 mutations.

The effects of the specific inhibitors is surprisingly modest, with or without NF2. The results are somewhat more robust in the spheroid system.

One of the most striking and potentially significant findings is that even the putative TEAD1-specific inhibitors seem to affect other TEADs, and that cross-reactivity may be required.

One issue I have is with the data and conclusions from Fig. 3D. Here, it is claimed that loss of TEAD4 leads to compensatory gain in TEAD1. This is evident in only one of two clones, and not seen in the reciprocal experiment with TEAD1 KOs. As this is a key point in the author's model, at a minimum, these immunoblots should be quantified.

Dear editor and reviewers,

Thank you for your time and the constructive feedback provided. We have edited the manuscript, accordingly. We have expanded our isogenic model, carried out additional experiments, and analyses to address critiques throughout the text and figures. We have also updated wordings at places.

We feel that the edits made have enhanced the legibility of the manuscript and have helped ensure the results are clearer and easier to interpret.

Below are the original reviewer comments in *italic* with our responses in **red**.

Reviewer: 1

The manuscript describes several cell-based assays to address the effects of inhibitors of the YAP/TAZ pathway. They provide results from 2D and 3D culture models using genetically engineered cell lines for different components of the YAP/TAZ-TEAD pathway. Cell paint technology is used to profile complex morphological phenotypes to distinguish YAP/TAZ-TEAD dependent effects from off-target YAP/TAZ-TEAD-independent effects. From this work the authors conclude that this pipeline of experimental models may help expedite the discovery of therapeutics effective in managing YAP/TAZ-TEAD driven mesothelioma, mitigating some of the risks of moving into in vivo assessment of inhibitor safety and efficacy.

The manuscript presents a large number of different experiments using different experimental models. However, it is difficult to appreciate the significance of the different results. For example, the data from experiments in 2D seem less relevant compared to those in 3D spheroid cultures. The interpretation is also complicated by the absence of dose-response curves. One would like to minimally see a dose-response curve for the observed phenotypes (e.g. Fig 1B). The exception is in figure S1B, but this presents a confusing result for the dose-response effects of the drugs used. In addition, in figure 3B, the dose response relation is very modest if at all present. It seems that IK930 has the most consistent effect, whereas in most other assays, this compound performed less in comparison to VT-107 and K-975.

We thank the reviewer for his/her feedback.

We agree that the experiments in 2D are less relevant than those performed using spheroid cultures and have added text to highlight this further. The use of 2D cultures was typically limited to initial validation assays and morphological analyses incompatible with 3D spheroid models. These data are relevant, as most drug development in general rely on 2D cell culture-based assays. We also elaborate on this aspect in the discussion. We have included additional discussion to highlight the advantage of the 3D model relative to 2D systems. Former figures have been revised and additional data generated to add dose-response curves for 2D viability (figure 2B) and YAP nuclear localisation (supplementary figure S1B). We have also added context to address instances where the dose-response relation of certain treatments is not clear, with a likely explanation being non-specific effects at high concentrations.

For many figures difficult to understand the way results presented. In many instances violin plots are used where bar charts would be more informative. It is also not clear what is depicted in these violin plots (replicates?, genes?) and the heterogeneity in the results seems very large and heterogeneous.

To enhance legibility, we have reformatted some violin plots to bar plots. However, violin plots were retained where quantifications are centred around 0, as we feel these are better suited to portray this data type. We have updated figure legends to ensure all data representations are detailed. We do agree that some of the results are heterogeneous, though high variance is a common characteristic of high content approaches, such as those included within the manuscript. To counter this, we include high replicate numbers permitted by these approaches, with the additional power provided to the statistical tests performed allowing for a high degree of confidence despite inherent variance.

The choice of drugs included in different experiments is inconsistent. For example, in figure 1C, IK-930 is missing, which is important for comparison with Fig 1B. In figure S1A, Dasatinib not included.

Thank you. We have updated the gene signature quantification in MeT-5A cells with missing compound for completeness. We have moved Fig 1C to supplementary figures and incorporated these data into Supplementary figure 1A. We have not prioritized the completion of the full compounds for HEK293As, as we mainly use this cellular model in this manuscript for mechanistic insights.

How should we interpret the difference between parental and TEAD knock-out with respect to the Yap/Taz signature genes? The way the data is presented is confusing. For example, when WT is compared to TEAD or NF2 knock-outs, there is no DMSO control.

We have performed qPCR analysis of YAP/TAZ signature genes in NF2/TEAD dKO spheroids and included this in the text. We have removed DMSO control data across figures, in which quantification is relative to DMSO control treated cells, this was inconsistently included in the previous version. We thank the reviewer for pointing this out to us.

The effect of perturbation of YAP or TEAD in spheroids with NF2-loss is presented, such experiments are not performed in 2D cell viability experiments.

These experiments have now been performed (supplementary figure S3A), and we have included discussions within the updated text.

The study presents perturbation effects of YAP or TEAD in spheroids with NF2 loss but does not include similar experiments in 2D cell viability assays. How does this relate to gene-expression signatures when comparing 2D to 3D (figure 1)? The conclusion that "These findings collectively point to the spheroid formation assay as effective to evaluate and probe YAP/TAZ-TEAD driven tumorigenic capacity" would be stronger if additional data were provided to demonstrate pathway activation/ inactivation in 3D cultures.

We have carried out qPCR analysis of YAP/TAZ signature genes in NF2/TEAD and NF2/YAP dKO spheroids (supplementary figure S3D). We agree that these additional data strengthen the conclusions.

Figure 2H is unclear in its implications. It should be represented in a manner similar to Figure 2F, where NF2 loss leads to increased growth that is mitigated by TEAD/YAP/TAZ inhibition.

Figures 3B and 3C also suffer from unclear data representation and should be revised for better interpretability.

We agree that the plotting of relative sensitivity rather than relative spheroid area on treatment may mask the full extent of treatment effects and makes direct comparisons between KO genotypes more difficult. However, as the manuscript is centred on the relative effectiveness of treatments for potential future clinical use, we believe that sensitivity is a more meaningful metric within this context and so these figures have been retained. We point to previously included figures that address the reviewer's concerns (supplementary figure S2E), and we have included additional figures (supplementary figure S3C) detailing relative areas for readers that may want to make additional direct comparisons.

The statement: "Considered together, these findings suggest that NF2-loss may result in a resistance to inhibition of YAP/TAZ-TEAD via activation of the Hippo pathway's core kinase cascade, while leaving NF2-deficient cells similarly vulnerable to TEAD inhibition as compared to those with intact NF2" is difficult to follow. The authors should clarify and refine this conclusion.

The statement on page 8: "Recurrent phenotype associated with off-target effects of YAP/TAZ-TEAD inhibition" appears incorrect. The small molecules used in the study have off-target effects that are independent of YAP/TAZ-TEAD inhibition.

We thank the reviewer for their feedback. These sentences have now been revised.

Reviewer: 2

This paper uses a standardized isogenic cell system to systematically evaluate hippo pathway inhibitors, particularly those that target TEAD proteins. The strength of the manuscript is in the model system, which eliminates many confounding genetic variables, and the use of 3D spheroid models, which accentuate the effects of the inhibitors. This is potentially significant, as TEAD inhibitors have entered clinical trials in mesothelioma and other cancers with NF2 mutations.

We thank the reviewer for their comments and appreciate the noted strength of the model system.

The effects of the specific inhibitors is surprisingly modest, with or without NF2. The results are somewhat more robust in the spheroid system.

This is a good observation, and we have included some points in the discussion to highlight the advantage of the 3D model relative to 2D systems for additional context.

One of the most striking and potentially significant findings is that even the putative TEAD1-specific inhibitors seem to affect other TEADs, and that cross-reactivity may be required.

We thank the reviewer for this comment and agree these are substantial observations.

One issue I have is with the data and conclusions from Fig. 3D. Here, it is claimed that loss of TEAD4 leads to compensatory gain in TEAD1. This is evident in only one of two clones, and not seen in the reciprocal experiment with TEAD1 KOs. As this is a key point in the author's model, at a minimum, these immunoblots should be quantified.

We thank the reviewer for the feedback. We have performed additional experiments and carried out requantifications. These combined data are significant across both clones (figure 3E).

We extend our thanks to the reviewers for the time they have given to their constructive feedback. We believe we have now addressed critiques raised through supplementary experiments and sweeping revisions to the text. On the back of these edits, the reviewer's feedback has greatly improved the legibility manuscript.

We very much hope that the manuscript is now ready for publication.

Dr Carsten Gram Hansen

The University of Edinburgh

July 7, 2025

RE: Life Science Alliance Manuscript #LSA-2025-03241-TR

Dr. Carsten G Hansen
University of Edinburgh
Center for Inflammation Research
Institute for Regeneration and Repair
Edinburgh BioQuarter, 4-5 Little France Drive
Edinburg EH16 4UU
United Kingdom

Dear Dr. Hansen,

Thank you for submitting your revised manuscript entitled "Pipeline to evaluate YAP-TEAD inhibitor potential - TEAD inhibition repress NF2-mutant mesothelioma". As you will see, the reviewers are satisfied with the changes in place. We would be happy to publish your paper in Life Science Alliance pending final revisions necessary to meet our formatting guidelines.

- Please add the X and Bluesky handles of your host institute/organization as well as your own or/and one of the authors in our system.
- The titles in both the system and the manuscript file must be consistent with each other. We suggest the following title: "Pipeline to evaluate YAP-TEAD inhibitors indicates TEAD inhibition represses NF2-mutant mesothelioma."
- Please be sure that the authorship listing and order are correct and match between the system and the manuscript file.
- Please consult our manuscript preparation guidelines <https://www.life-science-alliance.org/manuscript-prep> and make sure your manuscript sections are in the correct order.
- Please remove the 200-word summary file.
- The contributions selected for Adriano G. Rossi do not qualify them for authorship. Please either update the contributions in our system and the Author Contributions section of the manuscript, or let us know if the author needs to be removed (and potentially to the acknowledgment section).
- Please move your main and supplementary figure legends in the main manuscript text after the references section.
- Please add a Conflict of Interest statement to your main manuscript text.
- Please add a Data Availability statement to your main manuscript text. The Data Availability statement should address all data underlying the research presented in the manuscript. Please use the following guidelines:

Data in a public, open access repository

Include the repository name and persistent identifier (DOI, accession number, or permanent URL). All publicly available data should also be cited in the text and included in the reference list.

Data available upon reasonable request

We discourage making data available upon request rather than having it publicly available, but if there is no alternative for privacy or other reasons, describe what the data are and why they are not public, whom to contact (with a public email address), and conditions for re-use.

Data included in the article or supplementary data

We encourage authors to use a public repository rather than supplementary data.

A. FINAL FILES:

B. MANUSCRIPT ORGANIZATION AND FORMATTING:

Sincerely,

Reviewer #1 (Comments to the Authors (Required)):

The authors have addressed the comments and suggestions raised for the first version of this manuscript. The additional figures, the correct representation of the data and the additional discussion has improved the manuscript. I have no further comments or suggestions for this work. I do like to mention the rather small and heterogenous effects of the different specific inhibitors in this work, questioning the clinical translatability of the inhibitors used in this work.

Reviewer #2 (Comments to the Authors (Required)):

The authors have addressed my concerns. The results represent a modest advance, but an advance nevertheless and may help clarify results of other papers that use TEAD inhibitors

July 15, 2025

RE: Life Science Alliance Manuscript #LSA-2025-03241-TRR

Dr. Carsten Gram Hansen
University of Edinburgh
Center for Inflammation Research
Institute for Regeneration and Repair
Edinburgh BioQuarter, 4-5 Little France Drive
Edinburg EH16 4UU
United Kingdom

Dear Dr. Hansen,

Thank you for submitting your Research Article entitled "Pipeline to evaluate YAP-TEAD inhibitors indicates TEAD inhibition represses NF2-mutant mesothelioma". It is a pleasure to let you know that your manuscript is now accepted for publication in Life Science Alliance. Congratulations on this interesting work.

DISTRIBUTION OF MATERIALS:

Again, congratulations on a very nice paper. I hope you found the review process to be constructive and are pleased with how the manuscript was handled editorially. We look forward to future exciting submissions from your lab.

Sincerely,
